# Navoximod modulates local HSV-1 replication to reshape tumor immune microenvironment for enhanced immunotherapy via an injectable hydrogel

Qiuyu Zhuang[1,2,3,6], Binyu Zhao[1,3,6], Zhiwen Lin[1,3], Yuzhi Liang[1,3], Qingfu Zhao[1,3], Yunhao Wang[1,3], Naishun Liao[1,3], Haibin Tu[1,3], Youshi Zheng[1,2,3], Hengkai Chen[3,4], Yongyi Zeng[1,3,4], Da Zhang [1,2,3✉] & Xiaolong Liu [1,2,3,5✉]

Oncolytic virotherapy can lead to tumor lysis and systemic anti-tumor immunity, but the therapeutic potential in humans is limited due to the impaired virus replication and the insufficient ability to overcome the immunosuppressive tumor microenvironment (TME). To solve the above problems, we identified that Indoleamine 2, 3-dioxygenase 1 (IDO1) inhibitor Navoximod promoted herpes simplex virus type 1 (HSV-1) replication and HSV-1-mediated oncolysis in tumor cells, making it a promising combination modality with HSV-1-based virotherapy. Thus, we loaded HSV-1 and Navoximod together in an injectable and bio-compatible hydrogel (V-Navo@gel) for hepatocellular carcinoma (HCC) virotherapy. The hydrogel formed a local delivery reservoir to maximize the viral replication and distribution at the tumor site with a single-dose injection. Notably, V-Navo@gel improved the disease-free survival time of HCC- bearing mice and protects the mice against tumor recurrence. What's more, V-Navo@gel also showed an effective therapeutic efficacy in the rabbit orthotopic liver cancer model. Mechanistically, we further discovered that our combination strategy entirely reprogramed the TME through single-cell RNA sequencing. All these results collectively indicated that the combination of Navoximod with HSV-1 could boost the viral replication and reshape TME for tumor eradication through the hydrogel reservoir.

[1] The United Innovation of Mengchao Hepatobiliary Technology Key Laboratory of Fujian Province, Mengchao Hepatobiliary Hospital of Fujian Medical University, Fuzhou 350025, P. R. China. [2] Mengchao Med-X Center, Fuzhou University, Fuzhou 350116, P. R. China. [3] The Liver Center of Fujian Province, Fujian Medical University, Fuzhou 350025, P. R. China. [4] The First Affiliated Hospital of Fujian Medical University, Fuzhou 350025, P. R. China. [5] CAS Key Laboratory of Design and Assembly of Functional Nanostructures, Fujian Institute of Research on the Structure of Matter, Chinese Academy of Sciences, Fuzhou 350002, P. R. China. [6] These authors contributed equally: Qiuyu Zhuang, Binyu Zhao. ✉email: zdluoman1987@163.com; xiaoloong.liu@gmail.com

                    

In recent years, thanks to the success of immune checkpoint inhibitors (ICIs) and adoptive cell therapy, cancer immunotherapy (CIT) has achieved remarkable success in multiple cancer types[1]. As the fifth most common-occurring cancer globally, the immunotherapy for hepatocellular carcinoma (HCC) meets great obstacles of low response rate and undesirable clinical outcomes due to the complex immunological microenvironment with potent immunosuppressive effects[2–4]. Therefore, there is an urgent need to develop other immunotherapy strategies for HCC to overcome these obstacles. Oncolytic virotherapy has emerged as a promising therapy in many pre-clinical and clinical studies since the approval of T-VEC, a genetically modified herpes simplex virus type 1 (HSV-1), by U.S. Food and Drug Administration (FDA) for melanoma treatment in 2015[5,6]. Unlike immune checkpoint blockade which relies more on the modulation of the TME, oncolytic viruses not only impact on TME to induce anti-tumor immunity, but also enable a direct attack and killing against cancer cells[7]. Nevertheless, the durable remission rate of 16% in melanoma patients treated by T-VEC and the undesirable clinical outcomes in solid tumors raise urgent demands for further optimization of virotherapy[8,9]. The limited therapeutic effects are mainly caused by two reasons: the immunosuppressive tumor microenvironment and the anti-viral immunity by which the immune system clears the viral infection[10]. However, activation of anti-tumor immunity and the suppression of anti-viral immunity are competing in the TME and require careful modulation to keep the balance between them for an optimized virotherapy.

Indoleamine 2, 3-dioxygenase 1 (IDO1) is an important immunosuppressive protein which have been found to be upregulated in multiple types of tumors[11,12]. By catalyzing tryptophan (Trp) to kynurenine (Kyn), IDO1 inhibits the activities of CD8+ T cells and natural killer (NK) cells, improves the activation of regulatory T cells (Treg) and facilitates the recruitment of myeloid-derived suppressor cells (MDSCs)[13,14]. Accordingly, IDO1 inhibitors have been tested in several pre-clinical and clinical studies and demonstrated to be an effective strategy for immunotherapy[15,16]. In this study, we found that HSV-1 treatment could lead to up-regulation of IDO1 expression in HCC cells, which further acted as a negative feedback mechanism of the immune system to limit HSV-1 replication in tumor. Therefore, inhibiting IDO1 during HSV-1 virotherapy could play dual roles by modulating the immunosuppressive TME and enhancing viral replication in the tumor cells, which provided us with the rationale to combine IDO1 inhibitor Navoximod with HSV-1 oncolytic virotherapy for HCC treatment.

To achieve a local drug release and a maximized viral distribution at the tumor site, we encapsulated HSV-1 and Navoximod in injectable silk-hydrogels (referred to as V-Navo@gel) (Fig. 1). We demonstrated that the silk-hydrogels formed a local delivery reservoir, in which HSV-1 maintained a satisfactory cytotoxic ability and distribution at the tumor site and in the meantime attenuated the off-target damages to the peripheral organs. Our strategy resulted in complete therapeutic responses of HCC subcutaneous tumors, compared with only partial responses obtained with HSV-1 or HSV-1/Navoximod mixture without being loaded in the hydrogels. We further used single-cell RNA sequencing (scRNA-seq) to deeply profile the infiltration of immune cells to the tumor site, and revealed that V-Navo@gel could counter the immunosuppressive TME by remodeling immune cell populations dominated by accumulation of effector T cells and NK cells. The beneficial outcome could further expand to the orthotopic implanted VX-2 liver tumor model in the rabbit. Collectively, we demonstrated that the combination of Navoximod with HSV-1 augmented the efficacy of oncolytic virotherapy through the hydrogel reservoir, providing evidence to employ this strategy for clinical investigation.

## Results

### IDO1 inhibitor Navoximod promotes HSV-1 replication and HSV-1-mediated oncolysis.

Given that the clearance of HSV-1 by anti-viral immunity is one of the major obstacles for HSV-1-based virotherapy, we explored the genes inhibiting HSV-1 replication among interferon-stimulated genes (ISGs), which are the major participants of anti-viral immunity, for optimization of HSV-1 based virotherapy. We found that overexpression of IDO1, one of the reported ISGs[17], inhibited HSV-1 replication represented by the levels of envelope glycoprotein gD (Fig. 2a) and HSV-1 genomic DNA (which was indicated by gD and ICP47 DNA, Fig. 2b). Consistent results were obtained in the human HCC cell line SMMC-7721 and the mouse breast cancer cell line 4T1 that IDO1 overexpression inhibited HSV-1 replication, suggesting the universality of our findings (Supplementary Fig. 1a, b). We further used the GFP signal generated from GFP-tagged HSV-1 as another index of HSV-1 replication, and the GFP intensity in IDO1-overexpressing cells was observed to be much lower than that in wild type cells after HSV-1 infection (Fig. 2c). IDO1 has been reported to be overexpressed in multiple types of cancers[15], and we also confirmed it to be up-regulated in HCC tumor tissues according to The Cancer Genome Atlas (TCGA) RNA-Seq dataset and RNA-seq results of our group (Fig. 2d). Notably, qPCR and western blotting also revealed that HSV-1 treatment could induce the up-regulation of IDO1 expression in Hepa1-6 cells at both transcriptional and translational levels (Fig. 2e and Supplementary Fig. 1c). Thus, IDO1 was overexpressed in HCC tissues and could be further upregulated by HSV-1 treatment, which acted as a negative feedback mechanism of the immune system to limit HSV-1 replication in the tumor cells. Together with the previous reports that IDO1 is immunosuppressive against anti-tumor immunity, we believed that there was a necessity to block IDO1 during HSV-1-based virotherapy.

Navoximod is a highly selective IDO1 inhibitor. Based on the above findings that IDO1 could limit HSV-1 replication, we next investigated whether targeting IDO1 by Navoximod could reverse the inhibition of the HSV-1 replication and cytotoxic effects. Compared with HSV-1 treatment alone, HSV-1/Navoximod mixture (referred to as V-Navo) induced a dramatic enhancement of the genomic DNA level and the gD protein level of HSV-1, indicating that Navoximod improved HSV-1 replication (Fig. 2f, g). Consistently, we further validated that Navoximod improved HSV-1 replication in SMMC-7721 and 4T1 cells (Supplementary Fig. 1d, e). We also evaluated the effect of two additional IDO1 inhibitors Indoximod and Epacadostat on HSV-1 replication, and the results indicated that both compounds were capable to enhance HSV-1 replication in Hepa1-6 cells (Supplementary Fig. 1f). Moreover, HSV-1/Navoximod mixture decreased the cell viability of Hepa1-6 cells, indicating that Navoximod enhanced the HSV-1-mediated oncolysis of HCC cells (Fig. 2h). Furthermore, the subcutaneous HCC and 4T1 mice model indicated that the intratumoral injection of Navoximod augmented HSV-1 replication at the tumor site (Fig. 2i and Supplementary Fig. 1g). Collectively, these data demonstrated that Navoximod could strengthen the replication and oncolytic effects of HSV-1 both in vitro and in vivo, which suggested the potential and advantage of HSV-1 and Navoximod combinatorial virotherapy.

### Silk-hydrogels serve as a promising platform to deliver oncolytic virus.

To maximize viral distribution to tumor cells, we introduced silk-hydrogels generated from cocoons as the intratumoral delivery system for virus (Fig. 3a)[18]. The limited cytotoxicity and excellent biosafety of silk-hydrogels were first demonstrated in Hepa1-6 cells (Supplementary Fig. 2a). Next, we encapsulated HSV-1 into silk-hydrogels (Virus@gel) and verified

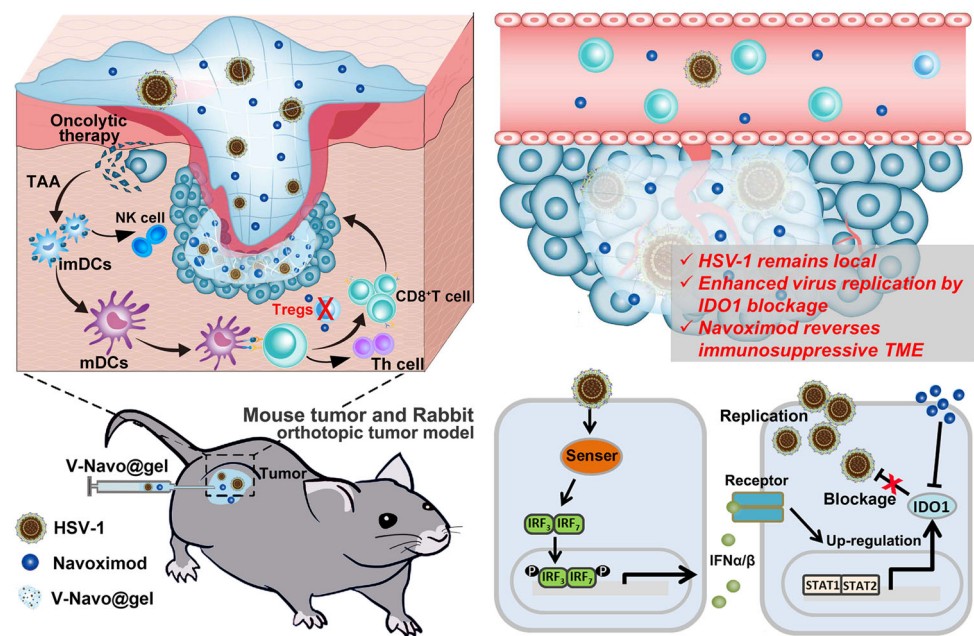

**Fig. 1 Schematic illustration of the mechanisms of TME reprogramming and virus replication modulation by HSV-1/Navoximod-loaded silk-hydrogels (V-Navo@gel) in the HCC model.** After intratumoral injection, hydrogels could limit HSV-1 at the tumor site to reduce the systemic toxicity. HSV-1 at the tumor site could induce the expression of the immunosuppressive protein IDO1, which impaired the therapeutic effect. Navoximod releasing from V-Navo@gel could therefore inhibit the enzymatic activity of IDO1, and subsequently enhance the virus replication and the antitumor immune responses at the same time.

its biological characteristics. Virus@gel showed injectable and gelatinous properties (Fig. 3b), which are important characteristics for a feasible hydrogel delivery system. The porous network structure of Virus@gel and the virion distribution inside hydrogels were verified by scanning electron microscopy and confocal microscopy, respectively (Fig. 3c, d and Supplementary Fig. 2b). Further evaluation of rheological and swelling properties demonstrated the gel state of Virus@gel and the low swelling ratio (Supplementary Fig. 2c, d), which indicated good mechanical properties and material stability.

We further demonstrated that the silk-hydrogels enabled to support the sustained release of the virions over 7 days (Fig. 3e, f, see Supplementary Fig. 2d). Cell viability assay demonstrated that Virus@gel caused 76.6% of cell death after coculturing with Hepa1-6 cells for 4 days, suggesting an excellent oncolytic effect of Virus@gel (Fig. 3g). Furthermore, the in vivo animal experiment showed that HSV-1 loaded in the silk-hydrogels could be detected 3 days after injection, while the signal of free HSV-1 weakened rapidly at the tumor site in 3 h, confirming that silk-hydrogels extend the localized retention of the virions at the tumor site (Fig. 3h). Consistently, Virus@gel enhanced the GFP signal of HSV-1 and the HSV-1 genomic DNA level at the tumor site compared with HSV-1 treatment alone (Fig. 3I, j). The above data demonstrated that our silk-hydrogels enabled to enhance the localized retention and replication of the virus at the tumor site.

We further demonstrated that silk-hydrogels prevented the virus diffusion and infection to healthy tissues, which was indicated by the HSV-1 genomic DNA level within the neural system, blood and peripheral organs after subcutaneous injection of HSV-1 or Virus@gel (Supplementary Fig. 3a, b). The blood biochemistry and routine indexes of mice indicated the limited toxicity of Virus@gel (Supplementary Fig. 3c). H&E images of the major organs consistently showed negligible damage among all the groups (Supplementary Fig. 3d).

Collectively, these results indicated that Virus@gel, which showed low systemic toxicity and good biosafety, could dramatically

maximize the virus distribution and replication at the tumor site, making it a potential delivery platform for virotherapy.

**V-Navo@gel inhibits primary tumor and tumor recurrence in HCC mouse model.** Based on the above findings, we next explored the anti-tumor effects of V-Navo@gel (HSV-1/Navoximod mixture encapsulated inside silk-hydrogels) in the subcutaneous HCC mouse model. C57BL/6 mice were inoculated subcutaneously with Hepa1-6 cells and treated with a single dose of PBS, HSV-1, HSV-1-loaded hydrogels (Virus@gel), Navoximod-loaded hydrogels (Navo@gel), HSV-1 plus Navoximod mixture (V-Navo) or V-Navo@gel, respectively (Fig. 4a). Among all the groups, mice treated with V-Navo@gel showed the strongest inhibitory effects on tumors, which was indicated by the change of tumor volume (Fig. 4b–d) and survival rate (Fig. 4e). Tumors in two out of six mice treated with V-Navo@gel were completely eradicated and all the mice in this group are survived 40 days after tumor inoculation without obvious body weight fluctuation (Fig. 4d–f). Consistent results were obtained in the 4T1 tumor model, in which V-Navo@gel treatment resulted in the strongest inhibitory effects on tumors growth without obvious body weight fluctuations (Supplementary Fig. 4). H&E, Ki67 and TUNEL staining of the tumor section indicated that V-Navo@gel caused the most serious cytolytic damages to tumor tissues among all the groups (Supplementary Fig. 5). These findings confirmed that V-Navo@gel could effectively prevent HCC tumor growth and induce oncolysis of tumor cells.

Next, a tumor re-challenge study was performed to determine whether V-Navo@gel could protect HCC cured mice from tumor recurrence (Fig. 4g). During the 22-day observation period, all the tumor-bearing mice re-challenged with Hepa1-6 cells in the V-Navo@gel group remained tumor-free (Fig. 4h–j). To further decipher the underlying mechanisms of the protection from tumor recurrence, we analyzed the abundance of naïve T cells ($CD62L^{high}CD44^{low}$), $T_{CM}$ ($CD62L^{high}CD44^{high}$) and $T_{EM}$

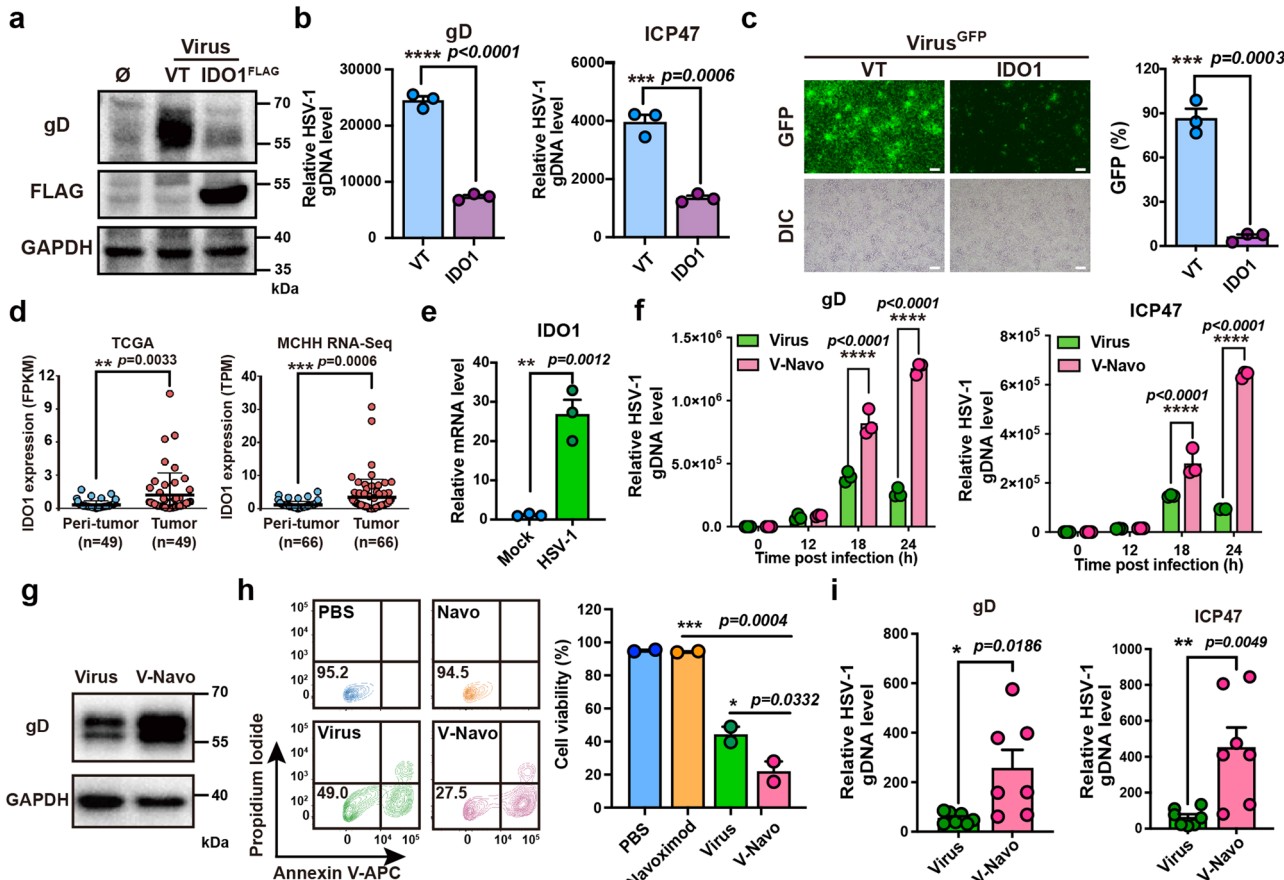

**Fig. 2 The combination effect of HSV-1 and IDO1 inhibitor Navoximod. a–c** Hepa1-6 cells transfected with empty vector (VT) or Flag-tagged IDO1 construct were infected with HSV-1 for 24 h. **a** Western blotting analysis of gD, FLAG and GAPDH. **b** RT-qPCR analysis of gD (left) and ICP47 (right) DNA level ($n = 3$; Data were shown as means ± SEM). **c** Representative fluorescence imaging (left) and flow cytometry analysis (right) of GFP-positive Hepa1-6 cells. Scale bar: 100 μm. $n = 3$. Data were shown as means ± SEM. **d** IDO1 expression in HCCs and their corresponding peri-tumor tissues was determined by TCGA HCC cohort (left) and transcriptome sequencing (right). Data were shown as means ± SEM. **e** Hepa1-6 cells infected with 20 MOI HSV-1 for 12 h were analyzed by RT-qPCR for IDO1 mRNA level ($n = 3$; Data were shown as means ± SEM). **f, g** Hepa1-6 cells were treated with 5 MOI HSV-1 or 5 MOI HSV-1 plus 1 μM Navoximod (V-Navo). **f** RT-qPCR analysis of gD (left) and ICP47 (right) DNA level at the indicated time points ($n = 3$; Data were shown as means ± SEM). **g** Western blotting analysis of gD and GAPDH. **h** Cell viability of Hepa1-6 cells with indicated treatments was determined by flow cytometry analysis using Annexin V-APC and PI staining ($n = 2$; Data were shown as means ± SEM). **i** RT-qPCR analysis of the intratumoral HSV-1 genomic DNA levels represented by gD (left) and ICP47 (right) 8 days after the indicated treatments ($n = 7$; Data were shown as means ± SEM).

(CD62L^low^CD44^high^) of both CD4^+^ and CD8^+^ T cells in the spleen of the mice after re-challenge experiments. In comparison with the control group, V-Navo@gel caused a decrease of the naïve T cells and $T_{CM}$ while an increase of $T_{EM}$ of both CD4^+^ and CD8^+^ T cells, suggesting that V-Navo@gel induced a conversion from naïve and central memory T cells to effector memory T cells for the rapid eradication of tumor recurrence (Fig. 4k).

**V-Navo@gel virotherapy remodels the immunosuppressive tumor microenvironment**. To conduct a more detailed and unbiased analysis of immune population and tumor-immune interaction in the tumors of V-Navo@gel-treated mice, PBS, Virus@gel or V-Navo@gel was injected intratumorally and the tumors were dissociated for single-cell RNA-sequencing (scRNA-seq) when there was a substantial tumor regression but no complete eradication. We first used unsupervised clustering data analysis to separate the total cells into distinct groups and populations of tumor cells, myeloid-derived cells, T cells, NK cells, fibroblasts, mural cells and fibroblasts were identified with distinct molecular signatures (Supplementary Fig. 6a, b). Among tumor-infiltrating immune cells, myeloid-derived cells and T/NK

cells are the predominant cell populations (Supplementary Fig. 6c), which were further classified based on the canonical markers for each population[19–21].

We then conducted unsupervised clustering of T/NK cells from scRNA-seq in control, Virus@gel and V-Navo@gel groups. A total of eight clusters emerged, including four clusters for CD8^+^ T cells, three clusters for CD4^+^ T cells and one cluster for NK cells, each with its unique signature genes (Fig. 5a, b). Comparison of these clusters among PBS, Virus@gel, and V-Navo@gel treatment was presented in Fig. 5c, d. Of note, the population of NK cells (cluster T_6) was enriched after V-Navo@gel treatment. V-Navo@gel also induced a great enrichment of cytotoxic CD8^+^ CTLs (cluster T_3, with overexpression of cytotoxic markers Gzmf/d/e and effector markers Ifng/Nkg7) and cycling CD8^+^ CTLs (cluster T_1 and T_2, with overexpression of cell cycle and DNA replication markers Cenpf/e, Top2a, Mki67, Npm1, and Rrm2). The dramatic increase in NK cells and CD8^+^ T cell subsets with functional cytotoxic and proliferating gene expression signatures explained the promising tumor eradication efficacy by V-Navo@gel treatment in the subcutaneous HCC mouse model. Meanwhile, helper T cells (cluster T_8), which are central for anti-tumor immune responses

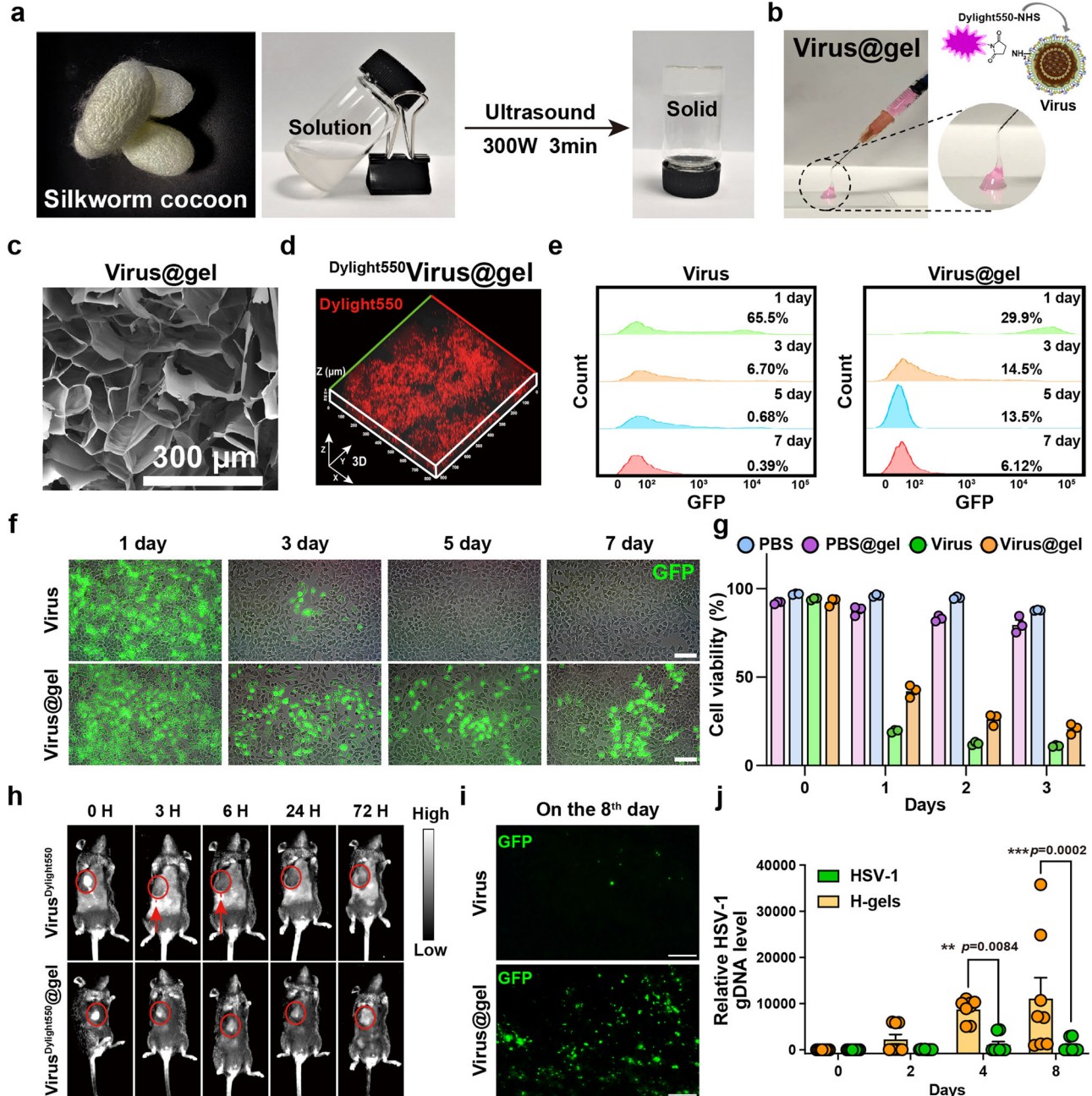

**Fig. 3 Preparation and characterization of HSV-1-based silk-hydrogels (Virus@gel). a** The image of the solution-to-gel process of silk-gels. **b** The image of HSV-1 encapsulated in the silk-hydrogels (Virus@gel) demonstrated the characteristics of gelation and injectability. **c** SEM image of Virus@gel. **d** 3D Confocal imaging of HSV-1 labeled with DyLight 550 inside silk-hydrogels. **e**, **f** The flow cytometry analysis (**e**) and the representative fluorescence images (**f**) of GFP-positive Hepa1-6 cells in the sustained release experiment. Scale bar: 100 μm. $n = 3$. Data were shown as means ± SEM. **g** The cell viability after PBS, PBS@gel, HSV-1, or Virus@gel treatment was determined by flow cytometry analysis using Annexin V-APC and PI staining ($n = 3$; Data were shown as means ± SEM). **h–j** Mice bearing Hepa1-6 tumors were intratumoral injected with HSV-1 or Virus@gel. **h** The accumulation of DyLight 550-labeled HSV-1 at the tumor site at the defined time points. Red circles indicated the tumor sites. Red arrows indicated the diffusion of the virus outside the tumors. **i** The representative fluorescence imaging of the frozen tumor sections 8 days after HSV-1 or Virus@gel intratumoral administration. GFP signals indicated cells infected by HSV-1. Scale bar: 100 μm. **j** RT-qPCR analysis of the intratumoral HSV-1 genomic DNA levels represented by gD at the defined time points ($n = 8$; Data were shown as means ± SEM).

by activating effector T cells and recruiting innate immune cells such as macrophages, were also enriched after V-Navo@gel treatment.

It has been established that myeloid populations in TME exhibit potent impacts on T cell immunity[22,23], thus we also focused on the variation of myeloid populations following V-Navo@gel virotherapy. The unsupervised clustering of myeloid-derived cells showed a total of eleven clusters, including seven for macrophages, one for monocytes and three for DCs (Fig. 5e, f). Notably, the population of cDC1 (cluster MD_10), which has been reported to prime both CD4$^+$ and CD8$^+$ T cells and directly enhance the T cell-mediated anti-tumor immunity[24], was enriched by V-Navo@gel treatment compared with PBS group and single Virus@gel treatment (Fig. 5g). In addition, we also

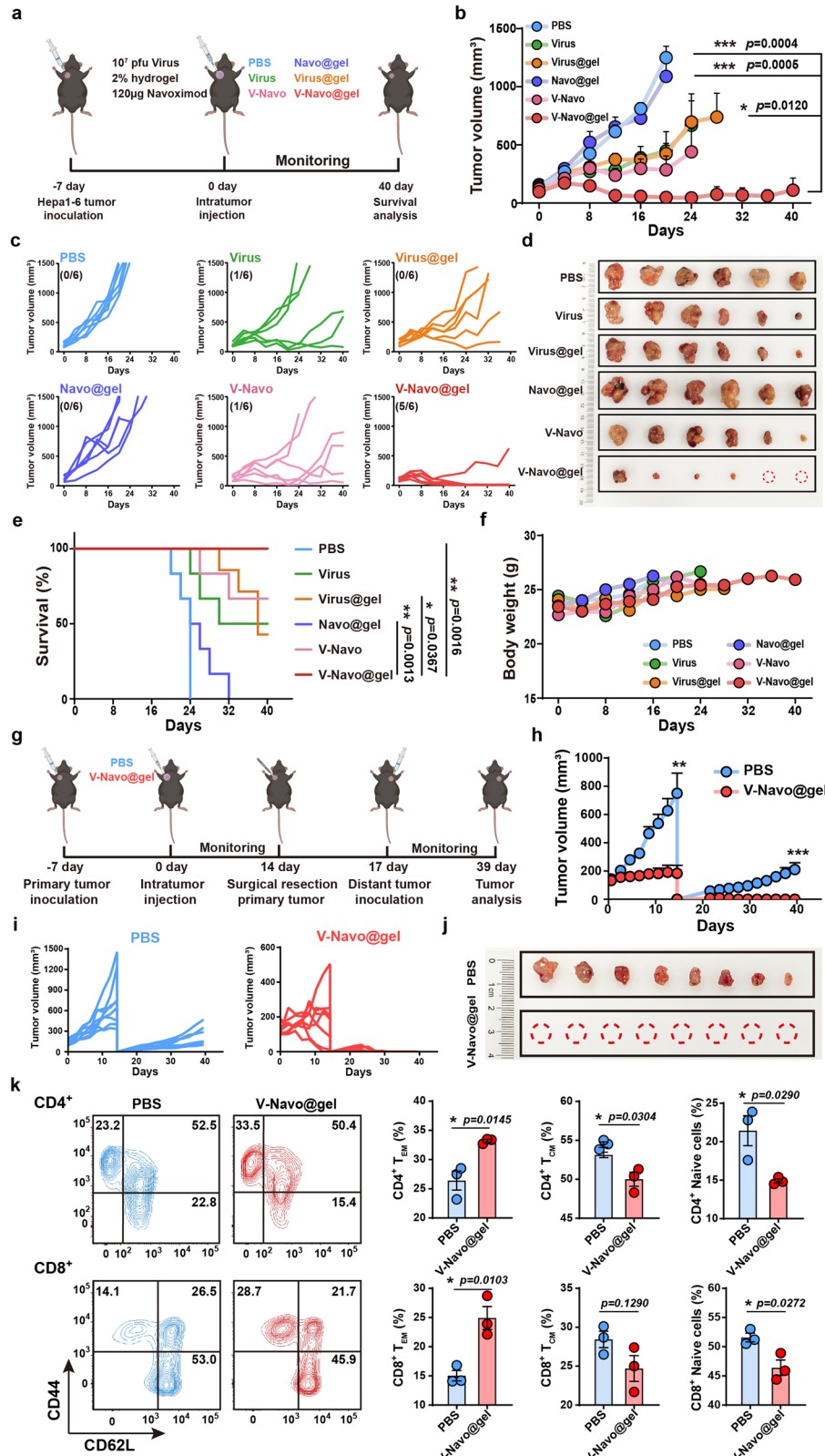

observed changes in the macrophage populations (Fig. 5g). Two functional phenotypes of macrophages M1 and M2 have long been established in which M1 macrophages are regarded as anti-tumor while M2 macrophages are contributed to pro-tumorigenic outcomes. V-Navo@gel caused an enrichment of cluster MD_3 macrophages with strong expression of M1 signature genes, which suggested that M1 macrophages also participated in the

anti-tumor process mediated by V-Navo@gel (Fig. 5g and Supplementary Fig. 6d). The above results collectively indicated that V-Navo@gel virotherapy elevated myeloid populations of cDC1 and M1 macrophages, and thus benefits anti-tumor efficacy.

To further assess the impact of the V-Navo@gel on biological activities of cancer cells, we performed GO analysis of the

**Fig. 4 In vivo elimination of subcutaneous HCC tumors by HSV-1 and Navoximod loaded silk-hydrogels (V-Navo@gel). a** Scheme of the therapeutic procedure. **b, c** Tumor volumes of mice with indicated treatments. The number in the brackets indicated the tumor objective response rate (ORR) of the mice in each group after the whole monitoring process ($n = 6$ mice per group; Data were shown as means ± SEM). **d** Tumor images isolated from mice after the indicated treatments. **e** The survival curves of mice with indicated treatments ($n = 6$; Data were shown as means ± SEM). **f** Inoculated mice weight changes during the whole measurement ($n = 6$; Data were shown as means ± SEM). **g** Scheme of the tumor rechallenge model. **h, i** Tumor volumes of mice with indicated treatments ($n = 8$; Data were shown as means ± SEM). **j** Tumor images isolated from mice after the indicated treatments. **k** Representative FACS plots (gated on $CD4^+$ or $CD8^+$ cells) and the percentage of $T_{em}$ ($CD62L^{low}CD44^{high}$), $T_{cm}$ ($CD62L^{high}CD44^{high}$) and naïve T cells ($CD62L^{high}CD44^{low}$) were examined by FACS ($n = 3$; Data were shown as means ± SEM).

differently expressed genes in cancer cells. GO analysis uncovered an enrichment of cell chemotaxis process by V-Navo@gel treatment, with upregulation of CCL and CXCL subfamilies which were previously recognized to modulate immune cell recruitment, differentiation, and expansion (Fig. 6a, b and Supplementary Fig. 7). To investigate the relation between upregulation of CCL/CXCL family genes and T cell-derived anti-tumor immunity, CellphoneDB was performed to analyze the communication of cancer cells and T cells. Ligand-receptor analysis revealed that CCL2/CCR2, CXCL10/CXCR3 and CXCL11/CXCR3 were the major interaction modules to mediate the crosstalk between cancer cells and different populations of T cells (Fig. 6c), and enhanced interactions of these ligand-receptor pairs between cancer cells and T cells were observed (Fig. 6d). Thus, the above data suggested that beside influencing immune-cell populations, V-Navo@gel treatment also impacted on cancer cells to modulate T cell recruitment to the tumor sites.

**V-Navo@gel induces the systemic modulation of anti-tumor immune responses in vivo.** We next performed flow cytometric analysis and IHC staining to confirm the systemic changes of the tumor immune microenvironment that we observed upon scRNA-seq (Fig. 7a). The maturation level of DCs ($CD11c^+/CD80^+/CD86^+$) in the tumor-draining lymph nodes was first analyzed by flow cytometry. Of note, V-Navo@gel triggered the highest level of DC maturation among all the groups (Fig. 7b). NK1.1 staining of the tumor sections demonstrated the increased intratumoral infiltration of NK cells after V-Navo@gel treatment, which is consistent with scRNA-seq analysis (Supplementary Fig. 8a). $CD8^+$ effector T cells in tumor tissues were also analyzed by flow cytometry and immunostaining, as dramatic changes in $CD8^+$ T cell population were observed upon scRNA-seq analysis after V-Navo@gel treatment. Consistently, both FACS and immunostaining indicated that V-Navo@gel effectively promoted the infiltration of $CD8^+$ T cells at the tumor site (Fig. 7c, d). Moreover, V-Navo@gel strongly inhibited the intratumoral infiltration of Tregs compared with the PBS group (Fig. 7e). Further analysis of the tumor lysates indicated that V-Navo@gel could elevate pro-inflammation cytokines (IL12 and IFNγ) and granzyme B, the marker of activated T cells, at the tumor site (Fig. 7f), indicating the elevated anti-tumor immune responses against tumors. We further performed an ex vivo IFNγ ELISPOT assay using splenic T cells to validate the generation of tumor specific T cells. Our ELISPOT assay showed that V-Navo@gel robustly activated antigen-specific $CD8^+$ T cells generated to against Hepa1-6 cells, indicating the successful generation of tumor-specific cytotoxic T lymphocytes. Nevertheless, we failed to observe the activation of tumor-specific $CD4^+$ T cells which mainly function in coordinating the immune response and require strong and sustained activation to produce IFNγ (Supplementary Fig. 8b). RT-PCR analysis also showed that, when compared to PBS or Virus@gel, V-Navo@gel induced an upregulation of CCL2, CXCL10 and CXCL11 (Supplementary Fig. 8c). This result supported the GO analysis and CellphoneDB analysis in Fig. 6, suggesting that our V-Navo@gel upregulated CCL/CXCL family genes for a more efficient T cell recruitment.

Overall, these findings suggested that V-Navo@gel comprehensively reprogrammed the tumor immune microenvironment for effective HCC virotherapy.

**In vivo anti-tumor activity of V-Navo@gel in rabbit VX-2 liver cancer model.** To further extend the clinical application potential, we examined the anti-tumor ability of V-Navo@gel in New Zealand white rabbits with implanted VX-2 tumor cell-induced carcinoma in the liver as the large animal model (Fig. 8a). We first demonstrated that HSV-1 could successfully infect and kill VX-2 tumor cells in vitro (Supplementary Fig. 9a). We further validated the association between IDO1 and HSV-1 in VX-2 tumor cells. As shown in Supplementary Fig. 9b and c, IDO1 overexpression strongly inhibited HSV-1 replication in VX2 cells, as indicated by the reduced levels of genomic DNA and the gD protein. Consistently, Navoximod improved HSV-1 replication in VX-2 cells, as indicated by increased levels of the genomic DNA and gD protein (Supplementary Fig. 9d, e). These findings confirmed the association between IDO and HSV-1 in VX-2 cells, and support the further in vivo studies in rabbit VX-2 cancer model. For the in vivo studies, VX-2 tumor tissues were minced and implanted into the exposed left lobe of the rabbit liver (Fig. 8b). After 12 days when the average volume of the tumors developed into 200 mm$^3$, the rabbits were randomized into PBS, V-Navo and V-Navo@gel treated groups, and the therapeutic agents were injected into the tumor site under ultrasound imaging guidance (Fig. 8c). Figure 8d showed the ultrasound images during the intervention. The weight of the animals was monitored every four days, and when the body weight loss of one group exceeded over 20%, all three groups were sacrificed, and the livers were isolated for assessing tumor size and location. One rabbit in the PBS-treated group died on Day 21. As shown in Fig. 8e, animals in PBS treated group exhibited the most obvious body weight loss (over 20%), while the body weight of animals in V-Navo and V-Navo@gel treated groups remained almost unchanged. At necropsy, rabbits treated with V-Navo@gel showed the strongest inhibitory effects on tumors compared with PBS and V-Navo groups (Fig. 8f, g). Collectively, these results indicated that the V-Navo@gel showed a strong anti-tumor effect in rabbit VX-2 liver cancer model.

**Discussion**

Oncolytic virotherapy have shown great potential in cancer immunotherapy. However, major challenges still exist, including the poor capacity for immune effector cells to infiltrate and function in the "cold" tumor immune environment, as well as the host developed antiviral strategies to limit the viral propagation. To find the solutions, several approaches have been reported including the incorporation of exogenous genes into the viral genomes and the combination with immune-checkpoint inhibitors or with adoptive transfer of immune effector cells[25,26]. Our study showed that we could overcome both obstacles by the combination of HSV-1 and IDO1 inhibitor Navoximod. It has been long recognized that IDO1 played opposite roles in pathogen infection by suppressing the replication of pathogens

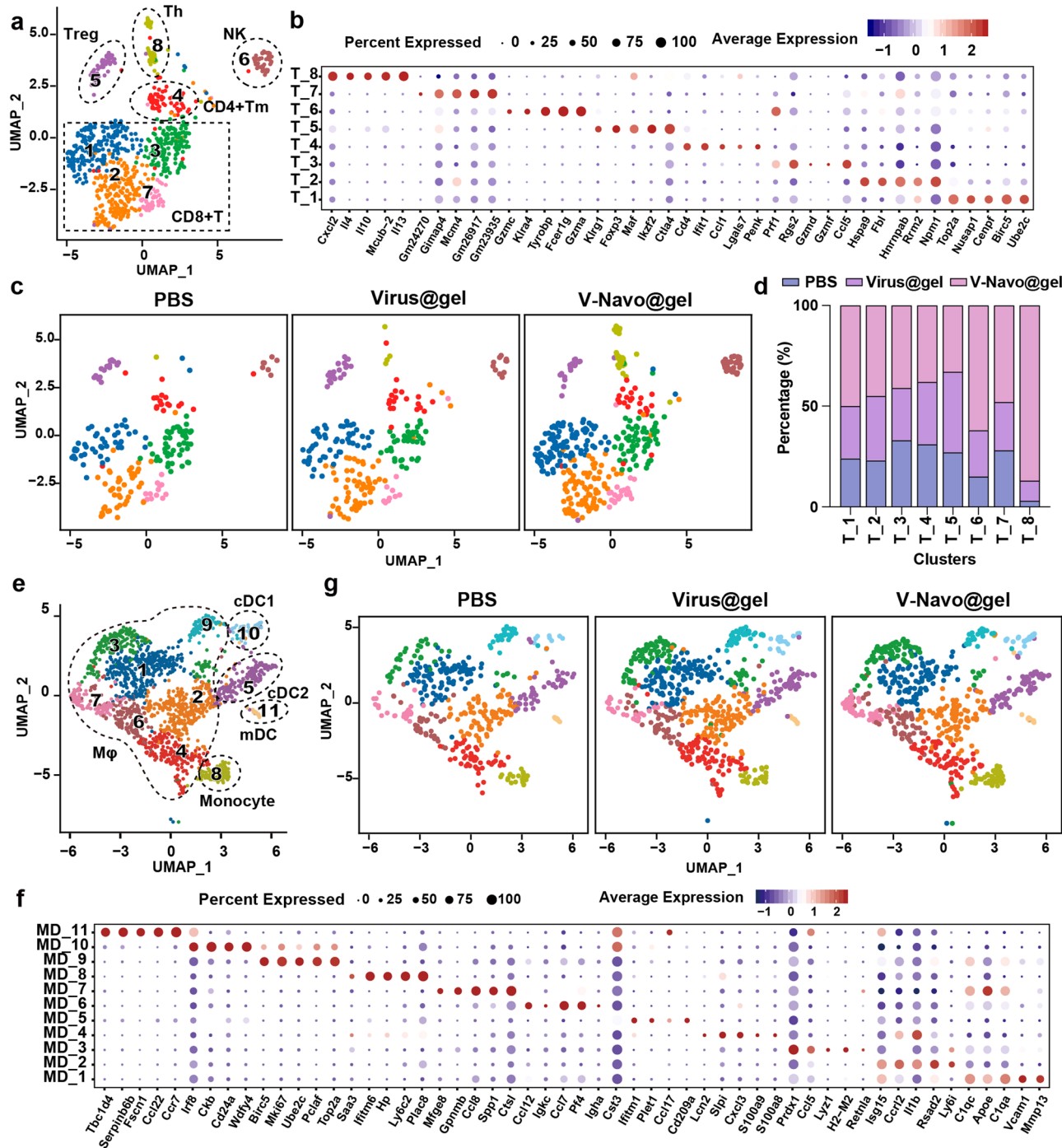

**Fig. 5 ScRNA-seq analysis revealed that V-Navo@gel induced an immunostimulatory reprogramming of TME. a** UMAP plot showing identified cell clusters within T/NK population. **b** Bubble heatmap showing expression of selected signature genes in each T/NK cluster. Bubble size represents the percentage of expressing cells, colored based on normalized expression levels. **c** Comparison of T/NK cell clusters between PBS, Virus@gel, and V-Navo@gel treatment. **d** Quantification of the proportion of each cell-type population based on treatment. **e** UMAP plot showing identified cell clusters within myeloid-derived population. **f** Bubble heatmap showing expression of selected signature genes in each myeloid-derived cluster. Bubble size represents the percentage of expressing cells, colored based on normalized expression levels. **g** Comparison of myeloid-derived cell clusters between PBS, Virus@gel, and V-Navo@gel treatment.

directly[27] or inhibiting T cell activity[28]. Recently, accumulating studies reported the immune moderating role of IDO1 in TME; it causes anergy of effector T cells and NK cells by inducing Trp depletion and promotes Treg differentiation by Kyn accumulation, which ultimately leads to tumor immune evasion[29–31]. In our study, we demonstrated that HSV-1 treatment on HCC cells upregulated IDO1 expression, and IDO1 impaired HSV-1

replication in HCC cells as a negative loop of the immune system to clear the viruses, raising the demand to inhibiting IDO1 during HSV-1-based virotherapy. The above findings provided us with the rational for a combination therapy of HSV-1 and IDO1 inhibitor. Another study also combined an oncolytic adenovirus carrying an activator of T cells with the IDO1 inhibitor for the treatment of brain tumors and achieve a promising therapeutic

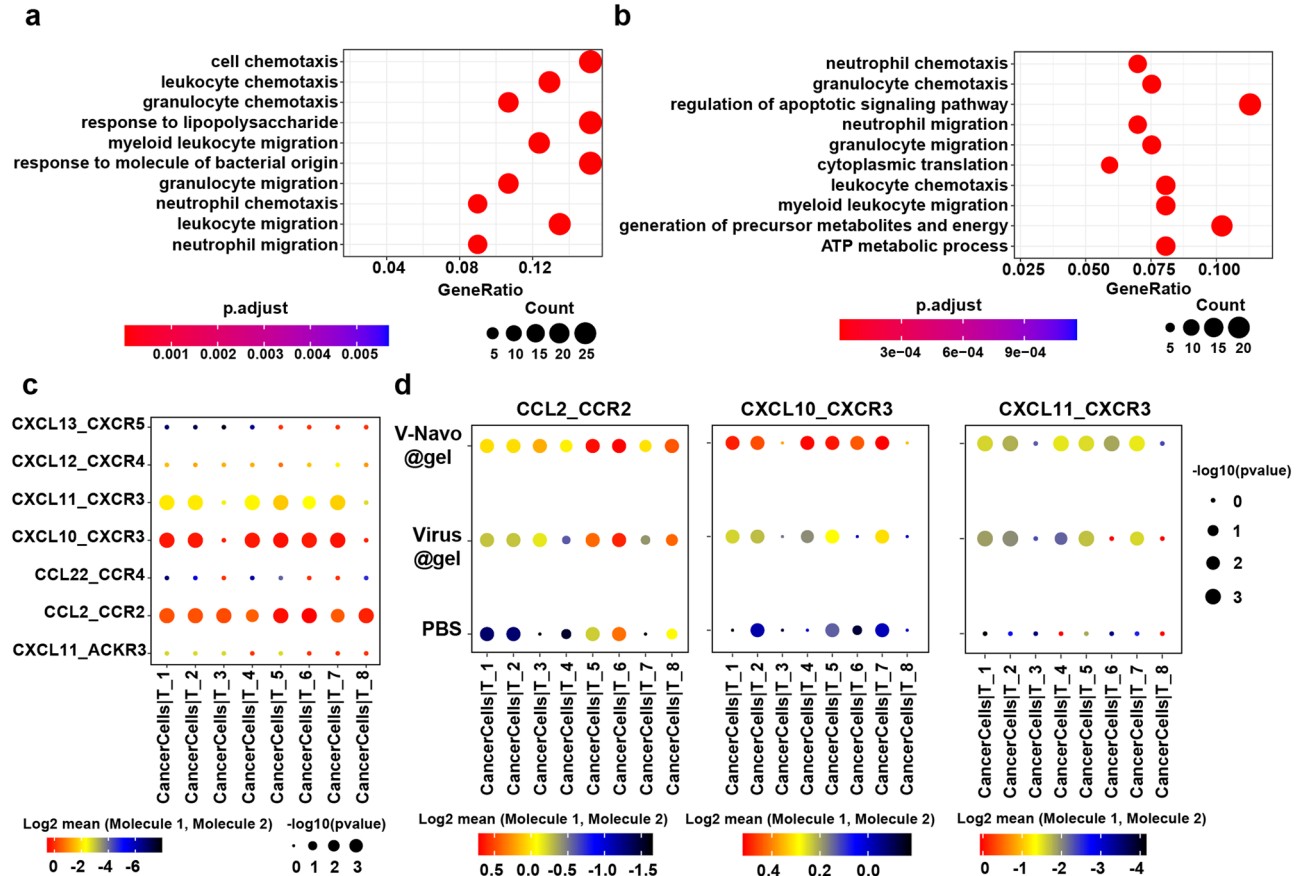

**Fig. 6 ScRNA-seq analysis revealed that V-Navo@gel treatment modulated communications between cancer cells and T cells. a**, **b** Top 10 terms in Gene Ontology (GO) analysis of differently expressed genes in V-Navo@gel versus PBS group (**a**) and V-Navo@gel versus Virus@gel group (**b**). **c** Overview of ligand-receptor interactions between cancer cells and different clusters of T cells. The bubble size represents p value. The color represents the means of the average expression level of interactions. **d** Selected ligand-receptor interactions between cancer cells and different clusters of T cells in the groups of PBS, Virus@gel and V-Navo@gel. The bubble size represents p value. The color represents the means of the average expression level of interactions.

outcome via reshaping the antitumor immune responses[32]. Therefore, we performed experiments to select the best inhibitors of IDO1 to induce HSV-1 replication, and found that Navoximod had the strongest effect compared to Indoximod or Epacadostat under each EC50 (Fig. 2f and Supplementary Fig. 1g). Additionally, the fact that Navoximod has already been successfully used in our group for cancer therapy also added to the confidence in our decision to choose Navoximod over other IDO-1 inhibitors for the HSV-1 combinatorial virotherapy[33]. Indeed, the animal models illustrated that the biocompatible hydrogel encapsulated with HSV-1 and Navoximod showed great therapeutic potential for both primary HCC and tumor recurrence.

Most clinical trials of oncolytic virotherapy adopt repeated dosing to maximize the viral distribution at the tumor site, which renders difficulties when intratumoral injection is required. Here we proposed a local delivery strategy to embed the viruses into silk-hydrogels. Silk-hydrogels is a novel biomaterial with extensive applications because of its excellent biocompatibility and have been used in the field of tissue engineering and drug delivery[34]. In this study, the hydrogels acted as a depot for the sustained release of the virions concentrated in the tumor tissues after a single-dose injection. Accumulating studies have reported that different types of hydrogels, including silk-elastin-like hydrogel, gelatin hydrogel, et al., have been utilized to encapsule adenoviruses for anti-tumor virotherapy[35–37]. We demonstrated that the in-situ silk-hydrogels we proposed here could

confine HSV-1 at the tumor site, which further induced the intratumoral virus replication and limited peripheral organ infection. Overall, this hydrogel system could achieve a sustainable release and enrichment of the virions at the tumor site, making it applicable to other oncolytic viruses as a universal delivery platform for virotherapy.

The analysis of single-cell RNA sequencing shed light on the immune populations that were predominant in the tumors treated with V-Navo@gel. Previous studies have indicated the boosting potential of oncolytic HSV-1 to trigger CD8+ T cells[38]. The scRNA sequencing together with the following FACS and immunofluorescence assay indicated that V-Navo@gel generated a strong CD8+ T cell responses, which explained the admirable therapeutic effects against tumors. As the dominant anti-tumor effector cells, NK cells were also enriched after V-Navo@gel treatment. We also found an increase in memory T cell population, which was consistent with the protection from tumor recurrence we observed. According to the scRNA-seq analysis and FACS, DC subsets were also modulated after V-Navo@gel treatment. It has been reported that cDC1 can present antigens directly to both CD4+ and CD8+ T cells and synergize with T cell immunity for an optimal anti-tumor efficacy[24]. We observed that V-Navo@gel induced the enrichment of cDC1 population, possibly due to the upregulated virus replication and oncolysis caused HSV-1 and Navoximod combination and subsequentially leading to the increased releasing of tumor antigens.

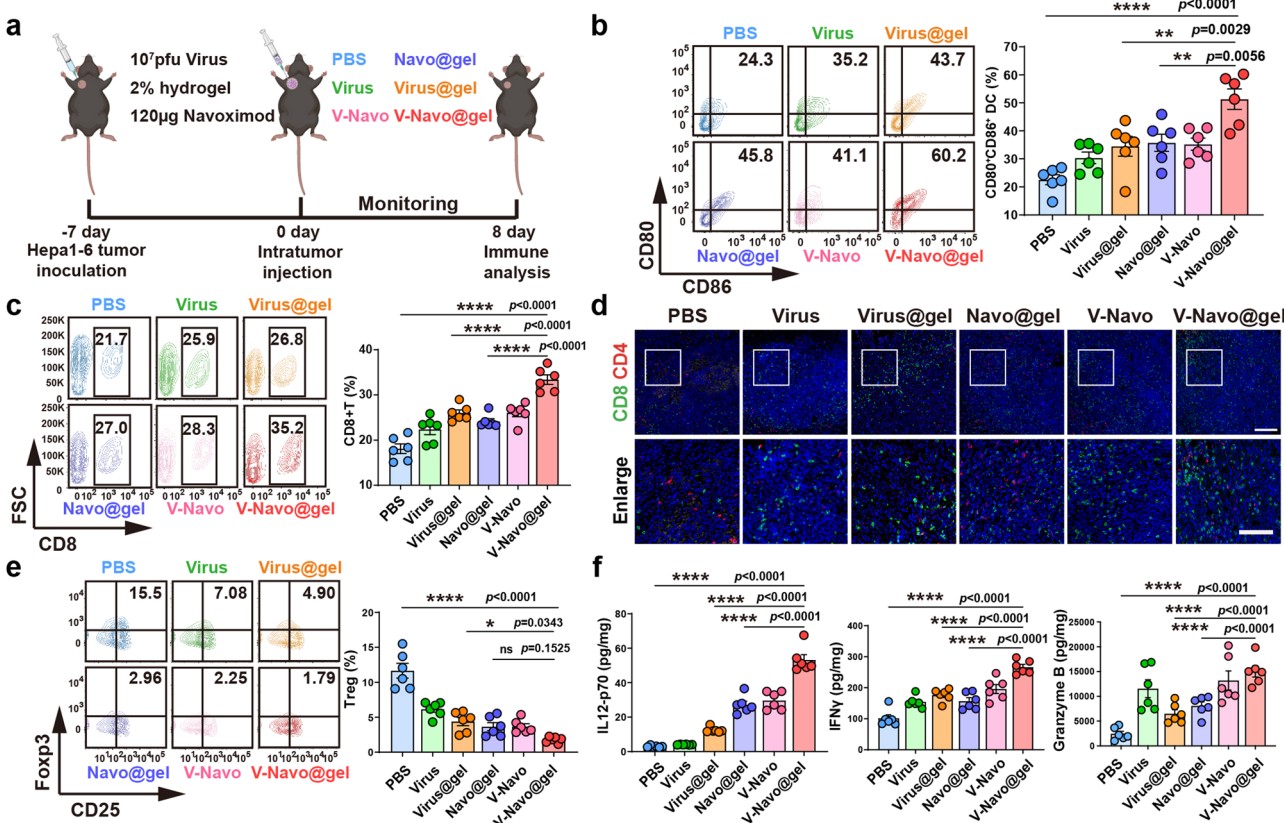

**Fig. 7 In vivo modulation of antitumor immune responses by V-Navo@gel. a** Scheme of immune response analysis during the therapeutic procedure. **b** Representative FACS plots (gated on CD11c+ DC cells) and the percentage of induced DC maturation in tumor-draining lymph nodes in tumor-bearing mice with indicated treatments (n = 6; Data were shown as means ± SEM). **c** Representative FACS plots (gated on CD3+ cells) and the percentage of CD8+T lymphocytes were examined by FACS (n = 6; Data were shown as means ± SEM). **d** Representative images of CD4 (red), CD8 (green) and DAPI (blue) staining of tumor sections after indicated treatments. Scale bar (top row): 200 μm. Scale bar (bottom row): 100 μm. **e** Representative FACS plots (gated on CD4+ cells) and the percentage of Tregs cells were examined by FACS (n = 6; Data were shown as means ± SEM). **f** Cytokine levels in tumor lysates by ELISA analysis after indicated treatments (n = 6; Data were shown as means ± SEM).

In addition to the immune cell populations, scRNA-seq further revealed that our combination therapy also reprogrammed the immunosuppressive cancer cells through the upregulation of chemotaxis pathways. V-Navo@gel treatment induced the expression of chemokines in cancer cells, including CCL2, CXCL10 and CXCL11 that drive the generation and recruitment of T cells, B cells and myeloid-derived cells[39]. While CCL2, CXCL10 and CXCL11 can directly recruit T cells into the tumors, CCL2 also enhances the recruitment of macrophages which subsequently increase intratumoral T cell infiltration[40–42]. Indeed, the cell-cell interaction analysis revealed that after V-Navo@gel treatment there were more communications between cancer cells and different clusters of T cells through CCL2/CCR2, CXCL10/CXCR3 and CXCL11/CXCR3 pairs, which might contribute to the increased intratumoral infiltration of T cells induced by V-Navo@gel. In all, the scRNA-seq analysis helped to reveal a reprogramming from the immunologically "cold" to "hot" tumors by V-Navo@gel treatment, which favors anti-tumor immunity.

Collectively, we demonstrated that HSV-1 and Navoximod could be delivered as a combination treatment through silk-hydrogels to facilitate the therapeutic efficacy of HCC by remodeling the immunosuppressive tumor microenvironment. Moreover, this strategy could be adapted to other immunotherapeutic modalities (including adoptive cells, immune checkpoint inhibitors, et al) as a localized reservoir for cancer therapy.

## Methods

**Cell culture.** Hepa1-6 and HEK293T cells were purchased from the American Type Culture Collection (ATCC). VX2, SMMC7721 and 4T1 cells were obtained from the Chinese Academy of Science (Shanghai, China). Vero cells were a gift from Jiahuai Han at Xiamen University. All the cells were cultured in Dulbecco's modified eagle medium (DMEM) supplemented with 10% (v/v) fetal bovine serum (ExCell, China). All cells were maintained at 37 °C and 5% CO$_2$. For transient transfection of IDO1 in the tumor cell lines, Lipofectamine 3000 (Thermo Fisher Scientific) was used following the manufacture's protocols.

**Virus propagation and titration.** GFP-HSV-1, which was generated by inserting GFP sequence into G47delta BAC, was used in all the experimental settings as oncolytic viruses and was propagated in Vero cells[43]. The titers of amplified viruses were determined on Vero cells using the viral plaque assay as previously described[44]. To determine HSV-1 genomic DNA levels, HSV-1 genomic DNA was extracted from tissue or cell culture samples infected with HSV-1 using TIANamp Genomic DNA Kit (TIANGEN, Germany). Then qPCR was performed for HSV-1 genomic DNA analysis with following primers:

HSV-1 gD: 5'-acgactggacggagattaca-3' and 5'-ggagggcgtacttacaggag-3';
HSV-1 ICP47: 5'-ggtgtggcacatcgaaga-3' and 5'-aacgggttaccggattacg-3'.

**Synthesis of silk-hydrogels with Navoximod and HSV-1 loading.** Bombyx mori fibrin solution was prepared according to the previous and our published works with slight modification[18,45]. Briefly, cocoons (5 g) were cut into pieces, boiled for 30 min in a Na$_2$CO$_3$ (20 mM) solution and then rinsed in ddH$_2$O to remove sericin proteins for three times. Extracted silk fibroin was then air-dried at 60 °C for 12 h. Afterwards, the dried silk fibroin was dissolved in 9.3 M LiBr salt solution at 60 °C for 4 h and then dialyzed (Mw, 3.5KDa) for 72 h to remove LiBr salt. The obtained silk fibroin solution was purified by centrifugation at 4 °C and then restored at 4 °C for further use. To obtain 2 wt% silk-hydrogels, gels were prepared by degumming 2 wt% silk solutions and sonicating them using an ultrasonic probe (Scientz-IID,

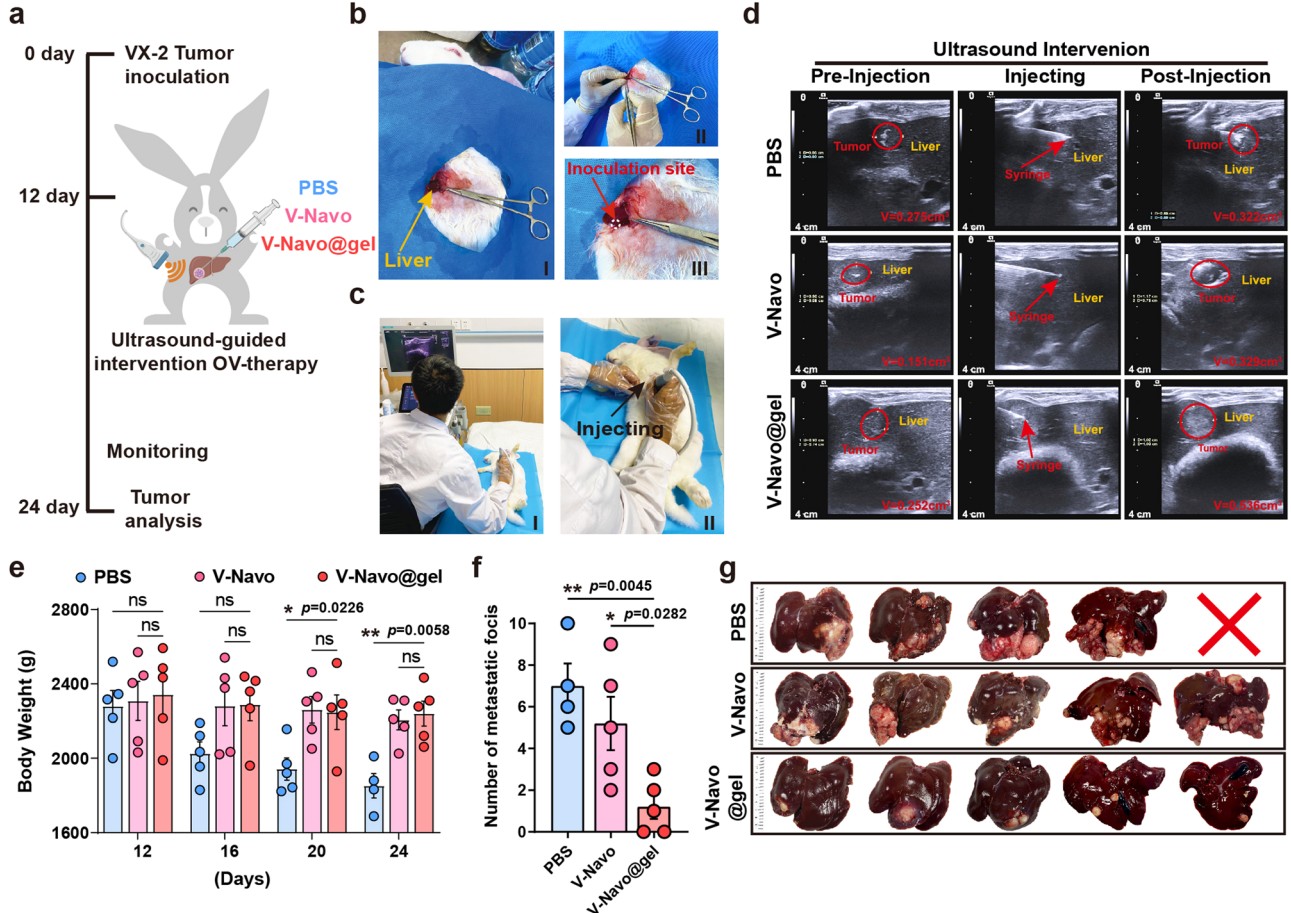

**Fig. 8 In vivo antitumor activity of V-Navo@gel in rabbit VX-2 liver cancer model. a** Scheme of the therapeutic procedure. **b** The implantation procedure of VX-2 tumor tissues into the exposed left lobe of the rabbit liver. **c** The injection procedure under the ultrasound guidance. **d** The ultrasound images during the intervention. **e** Weight changes of the inoculated rabbits during the whole measurement ($n = 5$; Data were shown as means ± SEM). **f** The number of metastatic nodules in the livers of rabbits with indicated treatments ($n = 4$ in PBS-treated group, $n = 5$ in V-Navo or V-Navo@gel-treated group; Data were shown as means ± SEM). **g** Images of livers isolated from rabbits with indicated treatments.

Ningbo Scientz Biotechnology, China) at 30% amplitude for 180 s. Then, the resulting solution was aging at 37 °C to obtain 2 wt% silk-hydrogels. Virus@gel or V-Navo@gel were prepared by mixing 2 wt% silk-hydrogels with the same volume of GFP-HSV-1 with the indicated virus titer or the same volume of solution containing HSV-1 and Navoximod.

**Evaluation of rheological and swelling properties**. For evaluating the rheological property, Virus@gel ($2 \times 10^6$ pfu) was placed in a Dynamic shear rheometer to perform rheology experiments. The storage modulus (G') and loss modulus (G") of Virus@gel were measured at appropriate strain and stress. For evaluation of swelling property, silk-hydrogels or Virus@gel were respectively placed in PBS, and the weight of each group was measured once every day. The swelling ratio was calculated by dividing this weight to the weight of initial gels.

**Sustained release analysis**. Virus@gel ($2 \times 10^6$ pfu) was placed in a cell strainer with an 8 μm pore size. The strainer was embedded in a 24-well plate cultured with $1 \times 10^5$ Hepa1-6 cells. Every 24 h, the strainer was washed twice with PBS and embedded in another 24-well plate with uninfected $1 \times 10^5$ Hepa1-6 cells, whereas the previous Hepa1-6 cells were subjected with fluorescence imaging and flow cytometry analysis to determine the percentage of GFP-positive cells.

**Subcutaneous murine cancer model**. For the establishment of subcutaneous Hepa1-6 tumors, an inoculum of $3 \times 10^6$ murine Hepa1-6 cells in 100 μL of sterile PBS was injected *s.c.* into the right armpit of 6-week-old female C57BL/6 mice. For the establishment of subcutaneous 4T1 tumors, an inoculum of $3 \times 10^6$ murine 4T1 cells in 100 μL of sterile PBS was injected *s.c.* into the right armpit of 6-week-old female Balb/c mice. When the tumors reached an average size of ~100 mm³, the mice were randomized into treatment groups by being intratumoral injected with PBS, $1 \times 10^7$ pfu HSV-1, $1 \times 10^7$ pfu HSV-1/120 μg Navoximod (V-Navo), $1 \times 10^7$ pfu GFP-HSV-1 encapsulated in 2% silk-hydrogels (Virus@gel), 120 μg Navoximod in 2% silk-gels (Navo@gel) or $1 \times 10^7$ pfu GFP-HSV-1/120 μg

Navoximod in 2% silk-hydrogels (V-Navo@gel), respectively. Tumor growth was monitored every two days and the tumor volume (V) was calculated using the following equation:

$$V = A * B^2/2$$

where A and B are the longer and shorter diameter (mm) of the tumor, respectively. The overall survival of the mice was monitored over 40 days and the end-point was determined when the tumor reached 1500 mm³.

For the establishment of tumor re-challenge model, HCC-bearing mice were prepared and administrated with PBS or V-Navo@gel according to the above-mentioned protocol. At day 14, surgical resection was performed to remove the primary tumor. At day 17, $3 \times 10^6$ Hepa1-6 cells in 100 μL of sterile PBS was injected *s.c.* into the opposite armpit of the mice. Afterward, the size of the re-challenge tumors was measured by caliper every 2 days, up to 40 days.

All the animal experiments were approved by the Institutional Review Board (IRB) of Mengchao Hepatobiliary of Fujian Medical University and were conducted in accordance with university guidelines.

**Tissue dissociation and library preparation for scRNA-seq**. Hepa1-6 tumors were treated with PBS, Virus@gel or V-Navo@gel ($1 \times 10^7$ PFU) intratumorally for 12 days. Then the tumors were dissociated and stored in the sCelLive™ Tissue Preservation Solution (Singleron Bio Com, Nanjing, China) on ice after the surgery within 30 min. Tumors were digested with 2 mL sCelLive™ Tissue Dissociation Solution (Singleron) by Singleron PythoN® Automated Tissue Dissociation System (Singleron) at 37 °C for 15 min. The solution was then centrifuged at 500 g for 5 mins and suspended softly with PBS. Then the single-cell suspensions ($1 \times 10^5$ cells/mL) were loaded into microfluidic devices using the Singleron Matrix® Single Cell Processing System (Singleron). Subsequently, the scRNA-seq libraries were constructed according to the protocol of the GEXSCOPE® Single Cell RNA Library Kits (Singleron)[46]. Individual libraries were diluted to 4 nM and

pooled for sequencing. At last, pools were sequenced on Illumina novaseq6000 with 150 bp paired end reads.

**scRNA-seq quantifications and statistical analysis**. Raw reads were processed with fastQC and fastP to remove low quality reads. Poly-A tails and adaptor sequences were removed by Cutadapt. After quality control, reads were mapped to the reference genome mus_musculus_ensembl_92 using STAR. Gene counts and UMI counts were acquired by featureCounts software. Expression matrix files for subsequent analyses were generated based on gene counts and UMI counts.

**Quality control, dimension-reduction and clustering**. Cells were filtered by gene counts between 200 to the top 2% gene counts and the top 2% UMI counts. Cells with over 20% mitochondrial content were removed, and 6000 cells per sample were randomly selected with the subset functions. We used functions from Seurat v3.1.2 for dimension-reduction and clustering[47]. All gene expression was normalized and scaled using NormalizeData and ScaleData. Top 2000 variable genes were selected by FindVariableFeautres for principal component analysis (PCA) analysis. Cells were separated by FindClusters using the top 20 principle components. UMAP algorithm was applied to visualize cells in a two-dimensional space.

**Cell type annotation**. Genes expressed in more than 10% of the cells in a cluster and with average log(Fold Change) of greater than 0.25 were selected as DEGs by Seurat v3.1.2 FindMarkers based on Wilcox likelihood-ratio test with default parameters. Then the cell type identity of each cluster was determined with the expression of canonical markers found in the DEGs according to SynEcoSys database or the previous literature.

**Cell-cell interaction analysis (CellPhoneDB)**. Cell-cell interaction were predicted based on known ligand–receptor pairs by Cellphone DB v2.1.0 (https://www.cellphonedb.org/). Permutation number for calculating the null distribution of average ligand-receptor pair expression in randomized cell identities was set to 1000. Individual ligand or receptor expression was thresholded by a cutoff based on the average log gene expression distribution for all genes across each cell type. Predicted interaction pairs with $p$ value < 0.05 and of average log expression >0.1 were considered as significant.

**Immune cell isolation and flow cytometry analysis**. For isolation and analysis of DCs in vivo, mice were sacrificed 8 days after treatment and tumor-draining lymph nodes were gently grinded and filtered through 40 μm filters. Then the cells were stained with the corresponding antibodies and examined by flow cytometry. The antibodies used for flow cytometry are listed below: Anti-CD11c-APC (eBioscience, USA); Anti-CD80-PE (eBioscience, USA); Anti-CD86-PE-Cy7 (eBioscience, USA). For isolation and analysis of TILs, mice were sacrificed 8 days after treatment and tumors were dissected, weighed, mechanically minced and treated with collagenase (1 mg/mL, Thermo Fisher Scientific, USA), hyaluronidase (0.2 mg/mL, Solarbio, China) and Dnase I (0.02 mg/mL, Sigma-Aldrich, USA) for 2 h at 37 °C with continuous agitation. Cells were then passed through a 40 μm filter and isolated by Ficoll-Paque density gradient centrifugation before stained with the corresponding antibodies and examined by flow cytometry. The antibodies used for flow cytometry are listed below: Anti-CD3-APC (eBioscience, USA), anti-CD4-FITC (eBioscience, USA) and anti-CD8-PE (eBioscience, USA) for TILs. Anti-CD25-PerCP-Cy5.5 (eBioscience, USA) and anti-Foxp3-PE-Cy7 (eBioscience, USA) for Tregs. For isolation and analysis of memory T cells, mice were sacrificed at the defined time point and spleens were dissected. Spleen cells were passed through a 40 μm filter and isolated by Ficoll-Paque density gradient centrifugation before stained with the corresponding antibodies and examined by flow cytometry. The antibodies used for flow cytometry are listed below: Anti-CD3-APC (eBioscience, USA), anti-CD4-FITC (eBioscience, USA), anti-CD8-PE (eBioscience, USA), anti-CD44-PE-Cy7 (eBioscience, USA) and anti-CD62L-PerCP-Cy5.5 (eBioscience, USA). For Enzyme-linked immunospot (ELISPOT) assay, mice were sacrificed 8 days after treatment, spleens were excised and CD8 + /CD4 + T cells were isolated via sorting. Afterwards, $3 \times 10^4$ sorted splenic T cells were cultured with $1.5 \times 10^5$ Hepa1-6 cells, and the resulting IFN-γ secretion were detected by ELISPOT kit (Mabtech, 3321-4APT-10) following the manufacture's instrument.

**Cytokine analysis for tumor lysate**. Tumors were isolated from mice on day 8 after treatment. 30 mg tissues were collected and lysed in RIPA lysis buffer (Beyotime Biotechnology, China) containing protease inhibitor cocktails (Med-ChemExpress, China). Tissues were homogenized with 5 mm magnetic beads at 60 Hz for 6 min and centrifuged at 16,000 g for 5 min at 4 °C. The supernatants were probed for IFNγ, Granzyme B or IL12p70 by ELISA (Boster Biological Technology, USA) according to the manufacturer's instructions.

**Histological evaluation**. Tumors were collected at sacrifice on day 8 after indicated treatments and stained with hematoxylin and eosin (H&E), Ki67 (R&D System, USA), TUNEL (R&D System, USA) and NK1.1 (ThermoFisher, USA) staining. Tumor sections were also subjected with immunofluorescence staining of

CD4 and CD8 by Servicebio, China. Major organs were collected on the same day and subjected with H&E staining.

**Orthotopic VX2 liver tumor model**. New Zealand White rabbits weighing approximately 2.5 kg were used. Frozen VX-2 tumor tissues were thawed and minced into 1–2 mm$^3$ sized pieces and implanted into the exposed left medial lobe of the liver. Ultrasound imaging was performed to monitor the propagation of the tumors. After 12 days when the average volumes of the tumors developed into 200 mm$^3$, animals were randomized into three groups and intratumorally injected respectively with PBS, $1 \times 10^9$ pfu GFP-HSV-1/2.5 mg Navoximod (V-Navo) or $1 \times 10^9$ pfu GFP-HSV-1/2.5 mg Navoximod in 2% silk-hydrogels (V-Navo@gel). The injections were performed under the ultrasound guidance. After treatment, the weight of the animals was monitored every four days, and when the loss of the body weight of animals from one group exceeded over 20%, all three groups were sacrificed and livers were dissociated.

**Western blot analysis**. Cell lysates were prepared in RIPA lysis buffer (Beyotime Biotechnology, China) containing PMSF and a protease inhibitor cocktail (Med-ChemExpress, China), and the protein content of the generated cell lysates was determined using the BCA protein assay (TransGen Biotech, China). Aliquots containing 30 μg of total protein were loaded on SDS-polyacrylamide gels and transferred to nitrocellulose membranes. After membranes were blocked with 5% BSA for 1 h, they were probed with indicated primary antibodies overnight at 4 °C, followed by incubation with the HRP-conjugated secondary antibodies (Abcam, 1:5000) for 1 h at room temperature. Primary antibodies were as follows: gD (21719, Santa Cruz, 1:500), DYKDDDDK-Tag (3P8, Abmart, 1:1000), GAPDH (AB0037, Abways, 1:10000), IDO1 (66528, Proteintech, 1:1000).

**Statistics and reproducibility**. The sample size was determined from the results of a preliminary study. The number used for each experiment is shown in figure legend. Multiple independent studies confirmed consistent results. Statistical analysis of data was analyzed through one-way or two-way of variance (ANOVA) for comparison among multiple groups or the two-tail paired Student's t-test for comparison between two groups. Data for survival was analyzed by the log-rank (Mantel-Cox) test. *$p < 0.05$ was set as statistically significant. **$p < 0.01$, ***$p < 0.001$, ****$p < 0.0001$. All the data were analyzed using GraphPad Prism and were shown as means ± SEM through at least three experiments.

**Reporting summary**. Further information on research design is available in the Nature Portfolio Reporting Summary linked to this article.

## Data availability

All data related to this study have been included in the article and its supplementary information. Source data for the graphs and charts in the figures is available as Supplementary Data. Images of original uncropped western blots and Gating strategies of FACS are included in Supplementary Fig. 10 and Supplementary Fig. 11. RNA-seq data were deposited into Genome Sequencing Achieve for Human database (https://ngdc.cncb.ac.cn/gsa-human/) under the accession number of HRA000464. ScRNA-seq data were deposited into Genome Sequencing Achieve database (https://hgdc.cncb.ac.cn/gsa) under the accession number of CRA008926. Other data are available from the corresponding author on reasonable request.

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

## Acknowledgements
This work was supported by the National Natural Science Foundation of China (Grant No. 62175031 to XL, and 82103317 to QZ); Fujian Province Outstanding Youth Science Fund Project (2022J06034 to DZ). Natural Science Foundation of Fujian Province of China (Grant No. 2020J02010 to XL, 2020J011173 to QZ); the Young and Middle-aged Talent Training Project of Fujian Provincial Health Commission (2020GGA073 to QZ); Joint Funds for the innovation of science and Technology of Fujian Province (Grant No. 2020Y9047 to QZ, 2021Y9216 to DZ); The Scientific Foundation of Fuzhou Municipal Health commission (Grant No. 2021-S-wp1 to YZ). We sincerely thank Weilin Liu from the College of Rehabilitation Medicine at Fujian University of Traditional Chinese Medicine for assisting the animal model building and diagnosis in our study.

## Author contributions
Q. Zhuang, D.Z. and X.L. designed the study. Q. Zhuang, B.Z., Z.L., Y.L., Q. Zhao, Y.W., N.L., H.T., Y.Z., D.Z. and H.C. conducted the experiments. Q. Zhuang, B.Z., Y.Z., D.Z. and X.L. analyzed the results. Q. Zhuang, B.Z., Y.Z., D.Z. and X.L. wrote and reviewed the manuscript. X.L. supervised the study.

## Competing interests
The authors declare no competing interests.
