## [Peer Review File · Communications Biology]

Reviewers' comments:

Reviewer #1 (Remarks to the Author):

In this manuscript, the authors developed an injectable and biocompatible silk-hydrogels for loading of both IDO1 inhibitor Navoximod and HSV-1 (V-Navo@gel). The synergistic effects between Navo and HSV-1 virotherapy were elaborately characterized. Such combination therapy was confirmed to against mouse HCC primary tumors, recurrence and even rabbit VX-2 liver tumors, by remodeling the immunosuppressive tumor microenvironment and generating immune memory effects. Overall, most of the experiments were elaborate and could greatly clarify the viewpoints, there are still little inadequacies should be considered and improved: 1). It is better to give the full names of the abbreviations in the "Abstract", like "HSV-1", "HCC", which would be more intelligible. 2). Some details in figures should be improved, such as Figure 7A showed "10 □ pfu virus", what is the meaning of "□"? And, IF images shown in the Figure 7D were not clear; the colors of the lines used in the groups "V-Navo" and "V-Navo@gel" were undistinguishable, please check and modify these questions.

Reviewer #2 (Remarks to the Author):

To briefly summarize,

This manuscript suggests a local, intratumor HSV-1 and Navoximod combination treatment employing injectable silk-hydrogel as a novel therapy for hepatocellular carcinoma. The in vitro data confirmed that the Navocimod reversed the inhibition of HSV-1 replication. Silk-hydrogel shown the sustained release of the HSV-1 and prevented the diffusion of HSV-1 to the normal tissues. This therapy exhibited excellent in vivo primary tumor and tumor recurrence inhibition with mouse and orthotopic rabbit models and verified that the anti-tumor responses were driven by the TME reprogramming, which eventually led to the induction of anti-tumor immunity.

Overall, I could not find significant issues or flaws that need to be addressed.

Novelty: I did find one paper that uses IDO inhibitor with virotherapy for cancer treatment(1). However, the concept of using injectable hydrogel to avoid systemic toxicity and maximize the anti-tumor effect was novel.

1. Nguyen TT, Shin DH, Sohoni S, Singh SK, Rivera-Molina Y, Jiang H, et al. Reshaping the tumor microenvironment with oncolytic viruses, positive regulation of the immune synapse, and blockade of the immunosuppressive oncometabolic circuitry. 2022;10(7):e004935.

Minor comments

1. Given that there are several IDO-1 inhibitors, the authors could elaborate in the introduction on why they have chosen Navoximod over other IDO-1 inhibitors.
2. Figure 2 (A) describes that the hepa1-6 cells have been transfected with empty vector and flag-tagged IDO construct. However, the cell transfection condition and the detail is not shown in the Method part.
3. Figure 2 (D) seems that the figure location has been mismatched with the legend. The figure with the header "TCGA" should go to the right side of the figure to match the explanation.
4. Low sample size (n=3) of the in vivo modulation of antitumor immune responses (Figure 7) and the orthotopic rabbit model (Figure 8) can be a issue. Usually, an animal study requires more than 5 of each group to prove the results. Moreover, The authors should perform power analyses based on anticipated results to estimate the required animal number per group to obtain a statistically significant difference.

Reviewer #3 (Remarks to the Author):

Comments:

1. The association between IDO and HSV-1 (oncolytic virus) is still controversial. Previous study also reported that HSV could inhibit IDO expression (Adv Virol. 2012) and another report showed

that IDO inhibition didn't influence HSV infection and replication (viral genome copies). Accord to the above mentioned studies, the authors should test more tumor lines (both mouse and human tumor lines) and in vivo tumor models, but not only Hepa1-6 for both in vitro and in vivo, which is very limited to get the conclusions to support clinical applications. In addition, additional IDO inhibitors should be tested.

2. In Fig 2-4, more tumor lines, not only Hepa1-6, both mouse and human lines should be tested. Otherwise, these discoveries maybe Hepa1-6 cell specific.

3. Fig 3, panel G, PBS gel and HSV-1 should be used as control.

4. Fig 4 shows that the antitumor effect of virus alone and virus@gel are similar, which is not constant with in vitro data in Fig 3, ould the authors explain these controversial data?

5. Fig 4, since the authors tried to argue the HSV-1 and IDO inhibitor combination could modulate the TME and enhance anti-tumor T cell responses, tumor specific CD8 and CD4 T cells should be determined by Elispots, but not only CD44 and CD62L with FACS.

6. In Fig 5 6 and 7, ScRNA-seq analysis results should be correlated with FACS data or RT-PCR data, the current data in Fig 7 are not associated with data in Fig 5 and 6. Again, tumor specific T cells should be determined in Fig 7.

7. Fig 7, the association between IDO and HSV-1 should be also tested with VX-2 tumor cell to confirm the same mechanisms of antitumor effect of HSV-1 and IDO inhibitor.

Dear Editor and Reviewers,

Thank you for carefully reviewing our manuscript entitled “**Navoximod modulates local HSV-1 replication to reshape tumor immune microenvironment for enhanced immunotherapy via an injectable hydrogel**” (Manuscript ID: COMMSBIO-22-3691-T) and providing constructive and insightful comments to improve our manuscript. Based on the insightful advice, we have revised the manuscript in terms of additional experimental studies, modifications of text and figures, and added extra discussions in the manuscript.

To facilitate your review of the revised manuscript, we marked the major changes of the manuscript in **red**. Please see below for our point-by point-response to all reviewers’ comments (*in italics*) below.

Reviewers' comments:

Reviewer #1 (Remarks to the Author):

In this manuscript, the authors developed an injectable and biocompatible silk-hydrogels for loading of both IDO1 inhibitor Navoximod and HSV-1 (V-Navo@gel). The synergistic effects between Navo and HSV-1 virotherapy were elaborately characterized. Such combination therapy was confirmed to against mouse HCC primary tumors, recurrence and even rabbit VX-2 liver tumors, by remodeling the immunosuppressive tumor microenvironment and generating immune memory effects. Overall, most of the experiments were elaborate and could greatly clarify the viewpoints, there are still little inadequacies should be considered and improved:

Response: We sincerely thank the reviewer for his / her careful reading and making the very useful comments. We also greatly appreciate the opportunity that we have been given to further revise the manuscript. We have carefully considered the comments and have revised the manuscript accordingly.

1). It is better to give the full names of the abbreviations in the “Abstract”, like “HSV-1”, “HCC”, which would be more intelligible.

Response: Thank you for your suggestion. Modifications have been made in the abstract of the manuscript.

2). Some details in figures should be improved, such as Figure 7A showed “10 □ pfu virus”, what is the meaning of “□”?

Response: Thank you very much. The error was attributable to the disparate format of Adobe Illustrator. It should be 10^7 pfu but not 10 □ pfu virus. We apologize for the mistake and have corrected it in Figure 7A of the revised version.

And, IF images shown in the Figure 7D were not clear; the colors of the lines used in the groups “V-Navo” and “V-Navo@gel” were undistinguishable, please check and modify these questions.

Response: Thank you very much. We rechecked the raw data and updated Figure 7D by replacing the previous images with higher resolution images with more distinguishable lines.

Reviewer #2 (Remarks to the Author):

To briefly summarize, This manuscript suggests a local, intratumor HSV-1 and Navoximod combination treatment employing injectable silk-hydrogel as a novel therapy for hepatocellular carcinoma. The in vitro data confirmed that the Navocimod reversed the inhibition of HSV-1 replication. Silk-hydrogel shown the sustained release of the HSV-1 and prevented the diffusion of HSV-1 to the normal tissues. This therapy exhibited excellent in vivo primary tumor and tumor recurrence inhibition with mouse and orthotopic rabbit models and verified that the anti-tumor responses were driven by the TME reprogramming, which eventually led to the induction of anti-tumor immunity. Overall, I could not find significant issues or flaws that need to be addressed.

Response: We sincerely thank the reviewer for his / her careful reading and making the very useful comments. We also greatly appreciate the opportunity that we have been given to further revise the manuscript. We have carefully considered the comments and have revised the manuscript accordingly.

Novelty: I did find one paper that uses IDO inhibitor with virotherapy for cancer treatment(1). However, the concept of using injectable hydrogel to avoid systemic toxicity and maximize the anti-tumor effect was novel.

1. Nguyen TT, Shin DH, Sohoni S, Singh SK, Rivera-Molina Y, Jiang H, et al. Reshaping the tumor microenvironment with oncolytic viruses, positive regulation of the immune synapse, and blockade of the immunosuppressive oncometabolic circuitry. 2022;10(7):e004935.

Response: Thanks very much for your suggestion, and we have cited this reference in the revised manuscript (page 11, line 283 to 286).

Minor comments

1. Given that there are several IDO-1 inhibitors, the authors could elaborate in the introduction on why they have chosen Navoximod over other IDO-1 inhibitors.

Response: Thanks very much for your suggestion. We performed experiments to select the best inhibitors of IDO1 to induce HSV-1 replication, and found that Navoximod had the strongest effect compared to Indoximod or Epacadostat under each EC50 24 hours after infection (Figure S1F and Figure R1). Therefore, we chose Navoximod in the further study. The corresponding description has been added in the first part of Results (page 4, line 95 to 98) as well as the Discussion part (page 11, line 286 to 291) of the revised manuscript.

Figure R1. (A) RT-qPCR analysis of gD (left) and ICP47 (right) DNA level at the indicated time points in Hepa1-6 cells treated with 5 MOI HSV-1 or 5 MOI HSV-1 plus 1 μ M Navoximod (V-Navo) (n=3; Data were shown as means \pm SEM). (B) RT-qPCR analysis of gD (left) and ICP47 (right) DNA level in Hepa1-6 cells treated with 5 MOI HSV-1, 5 MOI HSV-1 plus 1 μ M Indoximod (V-Indo) or 5 MOI HSV-1 plus 250 μ M Epacadostat (V-Epa) for 24 hours (n=3; Data were shown as means \pm SEM).

2. Figure 2 (A) describes that the hepa1-6 cells have been transfected with empty vector and flag-tagged IDO construct. However, the cell transfection condition and the detail is not shown in the Method part.

Response: Thanks very much for your consideration of this issue. For transient transfection of IDO1 in the tumor cell lines, Lipofectamine 3000 (Thermo Fisher Scientific) was used following the manufacture's protocols. The corresponding description has been added in the Method part of the revised manuscript on page 13, line 342 to 343.

3. Figure 2 (D) seems that the figure location has been mismatched with the legend. The figure with the header "TCGA" should go to the right side of the figure to match the explanation.

Response: Thanks very much for pointing this out. We apologize for this error and have revised it in the manuscript.

4. Low sample size (n=3) of the in vivo modulation of antitumor immune responses (Figure 7) and the orthotopic rabbit model (Figure 8) can be an issue. Usually, an animal study requires more than 5 of each group to prove the results. Moreover, The authors should perform power analyses based on anticipated results to estimate the required animal number per group to obtain a statistically significant difference.

Response: Thanks very much for your thoughtful comments on our manuscript. According to the reviewer's suggestion, we increased the sample size to n=6 in the immune modulation experiments (Figure 7B, C, E and F) and n=5 in the orthotopic rabbit model (Figure 8). The results were consistent with our previous results, and we have updated the corresponding figures in the revised manuscript.

Reviewer #3 (Remarks to the Author):

Comments:

1. The association between IDO and HSV-1 (oncolytic virus) is still controversial. Previous study also reported that HSV could inhibit IDO expression (Adv Virol. 2012) and another report showed that IDO inhibition didn't influence HSV infection and replication (viral genome copies). Accord to the above mentioned studies, the authors should test more tumor lines (both mouse and human tumor lines) and in vivo tumor models, but not only Hepa1-6 for both in vitro and in vivo, which is very limited to get the conclusions to support clinical applications.

Response: Thanks very much. According to the reviewer's suggestion, we extended our study to examine the effects of IDO1 on HSV-1 replication in the human HCC cell line SMMC-7721 and the mouse breast cancer cell line 4T1, respectively. Our results showed that IDO1 overexpression significantly inhibited HSV-1 replication, as evidenced by decreased levels of HSV-1 genomic DNA (indicated by gD DNA, see Figure S1A and Figure R2A) and gD protein (Figure S1B and Figure R2B). Consistently, IDO1 inhibitor Navoximod improved HSV-1 replication, as indicated by increased levels of the genomic DNA and gD protein in both SMMC-7721 and 4T1 cells (Figure S1D and E, Figure R2C and D). The *in vivo* 4T1 mice model also indicated that the intratumoral injection of Navoximod significantly augmented HSV-1 replication at the tumor site (Figure S1G and Figure R2E). The above results suggested that the relationship between IDO1 and HSV-1 replication that we observed was consistent and universal in different types of cancer cells in both human and mice. We have incorporated these new results into updated Figure S2. The corresponding description has been added in the revised manuscript on page 3, line 74 to 76 and page 4, lines 94 to 102.

Figure R2. (A and B) SMMC-7721 or 4T1 cells transfected with empty vector (VT) or IDO1 construct were infected with HSV-1 for 24 h. (A) RT-qPCR analysis of gD DNA level (n=3; Data were shown as means \pm SEM). (B) Western blotting analysis of gD and GAPDH. (C and D) SMMC-7721 or 4T1 cells were treated with 5 MOI HSV-1 or 5 MOI HSV-1 plus 1 μ M Navoximod (V-Navo). (C) RT-qPCR analysis of gD DNA level (n=3; Data were shown as means \pm SEM). (D) Western blotting analysis of gD and GAPDH. (E) RT-qPCR analysis of the intratumoral HSV-1 genomic DNA levels in 4T1 tumor model 8 days after the indicated treatments (n=5; Data were shown as means \pm SEM).

In addition, additional IDO inhibitors should be tested.

Response: Thanks very much for suggestion. According to the reviewer's suggestion, we further tested two additional IDO1 inhibitors Indoximod and Epacadostat, and found that both compounds were able to enhance HSV-1 replication in Hepa1-6 cells (Figure S1G and Figure R1). The corresponding description has been added in the revised manuscript on page 4, line 95 to 98.

2. In Fig 2-4, more tumor lines, not only Hepa1-6, both mouse and human lines should be tested. Otherwise, these discoveries maybe Hepa1-6 cell specific.

Response: Thanks very much for your attention to this problem. To determine if our findings were specific to Hepa1-6 cells, we first examined the impact of IDO1 overexpression or inhibition on HSV-1 replication *in vitro* using the human HCC cell line SMMC-7721 and the mouse breast cancer cell line 4T1, as described in our response to comment #1 and the updated Figure S1 of the revised manuscript. Consistent with the

results in Hepa1-6 cells, we observed that both IDO1 overexpression and IDO1 inhibition could influence the virus replication in SMMC-7721 and 4T1 cells. Therefore, we further validated the anti-tumor effect of V-Navo@gel in a 4T1 tumor-bearing mice. Specifically, BALB/c mice were inoculated subcutaneously with 4T1 cells and treated with a single dose of PBS, HSV-1, HSV-1-loaded hydrogels (Virus@gel), Navoximod-loaded hydrogels (Navo@gel), HSV-1 plus Navoximod mixture (V-Navo) or V-Navo@gel, respectively. Among all the groups, mice treated with V-Navo@gel showed the most significant inhibitory effects on tumor growth without significant body weight fluctuation (Figure S4 and Figure R3). These findings suggested that the improvement of HSV-1 replication by Navoximod is not cell-type specific and V-Navo@gel has the potential to be a universal anti-tumor therapy for both Hepa1-6 and 4T1 tumor models. The corresponding description has been added in the revised manuscript on page 6, line 146 to 148.

Figure R3. In vivo elimination of subcutaneous 4T1 tumors by HSV-1 and Navoximod loaded silk-hydrogels (V-Navo@gel). (**A and B**) Tumor volumes of mice with indicated treatments. ($n=5$; Data were shown as means \pm SEM). (**C**) Inoculated mice weight changes during the whole measurement ($n=5$; Data were shown as means \pm SEM). (**D**) Tumor images isolated from mice after the indicated treatments.

3. Fig 3, panel G, PBS gel and HSV-1 should be used as control.

Response: Thanks very much for your suggestion. According to the reviewer's suggestion, we carried out experiments in Figure 3G with the inclusion of PBS@gel and Virus as controls. The updated version of Figure 3G (see also Figure R4) can now be found in the revised manuscript.

Figure R4. The cell viability after PBS, PBS@gel, HSV-1, or Virus@gel treatment was determined by flow cytometry analysis using Annexin V-APC and PI staining (n=3; Data were shown as means \pm SEM).

4. Fig 4 shows that the antitumor effect of virus alone and virus@gel are similar, which is not constant with *in vitro* data in Fig 3, could the authors explain these controversial data?

Response: Thank you for bringing attention to this issue. In Figure 3H-J, it was suggested that silk-hydrogels enhance the localized retention and replication of the virus within the tumor. However, in Figures 3E-G, virus infection alone resulted in higher virus replication and more severe oncolytic effects at the initial time point post-infection. This discrepancy could be attributed to the delay in virus release due to the characteristics of silk-hydrogels. Thus, in the *in vivo* tumor model, relatively small tumors were attacked by higher doses of virions in the virus-only group at the initial time point. As a consequence, a comparable antitumor efficacy was observed between the virus-only treatment and virus@gel treatment, as demonstrated in Figure 4. Nevertheless, our study also demonstrated that silk-hydrogels effectively prevented virus diffusion and limited peripheral organ infection, as shown in Figure S3A. These findings indicate the potential of utilizing silk hydrogels as a delivery platform for virotherapy.

5. Fig 4, since the authors tried to argue the HSV-1 and IDO inhibitor combination could modulate the TME and enhance anti-tumor T cell responses, tumor specific CD8 and CD4 T cells should be determined by Elispots, but not only CD44 and CD62L with FACS.

Response: Thanks very much for your valuable suggestion. To validate the generation of tumor specific T cells, we performed an *ex vivo* interferon (IFN)- γ ELISPOT assay using splenic T cells from mice treated

with either PBS or V-Navo@gel 8 days after administration. Our ELISPOT assay showed that V-Navo@gel significantly activated antigen-specific CD8+T cells generated against Hepa1-6 cells, indicating the successful generation of tumor-specific cytotoxic T lymphocytes. However, we failed to observe the activation of tumor-specific CD4+T cells which mainly function in coordinating the immune response and require strong and sustained activation to produce IFN- γ . We have incorporated this result into Figure S8B (see also Figure R5) of our revised manuscript. The corresponding description has been added in the revised manuscript on page 9, line 231 to 237.

Figure R5. Representative images and analysis of the tumor specific T cells by IFN- γ ELISPOT assay (n=5; Data were shown as means \pm SEM).

6. In Fig 5 6 and 7, ScRNA-seq analysis results should be correlated with FACS data or RT-PCR data, the current data in Fig 7 are not associated with data in Fig 5 and 6. Again, tumor specific T cells should be determined in Fig 7.

Response: Thanks very much for your consideration of this issue. Both the scRNA-seq analysis (Figure 5) and the analysis of tumor immune microenvironment by FACS and immunostaining consistently showed that V-Navo@gel treatment induced a significant enrichment of NK cells, cytotoxic CD8+ CTLs and DC cells. According to the reviewer's suggestion, we further performed RT-PCR analysis in the *in vivo* Hepa1-6 tumor model to examine the expression level of CCL/CXCL family genes. RT-PCR analysis showed that, when compared to PBS or Virus@gel, V-Navo@gel induced a significant upregulation of CCL2, CXCL10 and CXCL11 (Figure S8C and Figure R6). This result supported the GO analysis and CellphoneDB analysis in Figure 6, suggesting that our V-Navo@gel upregulated CCL/CXCL family genes in the chemotaxis pathways for a more efficient T cell recruitment. We also performed an *ex vivo* interferon (IFN)- γ ELISPOT assay to determine the tumor specific T cells and the results showed that V-Navo@gel significantly generated tumor-specific cytotoxic T lymphocytes (see our response to comment #5, updated Figure S8B

and Figure R5). The corresponding description has been added in the revised manuscript on page 9, line 231 to 241.

Figure R6. RT-qPCR analysis of the intratumoral CCL2, CXCL10 and CXCL11 expression 8 days after the indicated treatments (n=5; Data were shown as means \pm SEM).

7. Fig 7, the association between IDO and HSV-1 should be also tested with VX-2 tumor cell to confirm the same mechanisms of antitumor effect of HSV-1 and IDO inhibitor.

Response: Thanks very much for your suggestion. According to the reviewer's suggestion, we have tested the effect of IDO1 overexpression and the IDO1 inhibitor on HSV-1 replication in VX-2 tumor cells. As shown in Fig S9B and C (see also Figure R7A and B), IDO1 overexpression strongly inhibited HSV-1 replication in VX2 cells, as indicated by the reduced levels of genomic DNA and the gD protein. Consistently, IDO1 inhibitor Navoximod improved HSV-1 replication in VX-2 cells, as indicated by increased levels of the genomic DNA and gD protein (Figure S9D and E, Figure R7C and D). These findings confirmed the association between IDO and HSV-1 in VX-2 cells, and further support the mechanism of antitumor effect of V-Navo@gel observed in our previous tumor model. We have incorporated this information in Figure S8 of our revised manuscript. The corresponding description has been added in the revised manuscript on page 9, line 247 to 253.

Figure R7. (A and B) VX-2 cells transfected with empty vector (VT) or Flag-tagged IDO1 construct were infected with HSV-1 for 24 h. (A) RT-qPCR analysis of gD DNA level (n=3; Data were shown as means \pm SEM). (B) Western blotting analysis of gD and GAPDH. (C and D) VX-2 cells were treated with 5 MOI HSV-1 or 5 MOI HSV-1 plus 1 μ M Navoximod (V-Navo). (C) RT-qPCR analysis of gD DNA level (n=3; Data were shown as means \pm SEM). (D) Western blotting analysis of gD and GAPDH.

REVIEWERS' COMMENTS:

Reviewer #2 (Remarks to the Author):

The authors clarified and addressed my minor concerns.

Reviewer #3 (Remarks to the Author):

According to the revisions, the manuscript can be accepted.